# Quantitative mechanistic model reveals key determinants of placental IgG transfer and informs prenatal immunization strategies

**Remziye E. Wessel[1], Sepideh Dolatshahi**[1,2]*

**1** Department of Biomedical Engineering, University of Virginia School of Medicine, Charlottesville, Virginia, United States of America, **2** Carter Immunology Center, University of Virginia School of Medicine, Charlottesville, Virginia, United States of America

* sdolatshahi@virginia.edu

**Data Availability Statement:** All data files and codes used to run the model can be found at our GitHub: https://github.com/Dolatshahi-Lab/Transplacental-antibody-transfer-model. Model

## Abstract

Transplacental antibody transfer is crucially important in shaping neonatal immunity. Recently, prenatal maternal immunization has been employed to boost pathogen-specific immunoglobulin G (IgG) transfer to the fetus. Multiple factors have been implicated in antibody transfer, but how these key regulators work together to elicit selective transfer is pertinent to engineering vaccines for mothers to optimally immunize their newborns. Here, we present the first quantitative mechanistic model to uncover the determinants of placental antibody transfer and inform personalized immunization approaches. We identified placental FcγRIIb expressed by endothelial cells as a limiting factor in receptor-mediated transfer, which plays a key role in promoting preferential transport of subclasses IgG1, IgG3, and IgG4, but not IgG2. Integrated computational modeling and *in vitro* experiments reveal that IgG subclass abundance, Fc receptor (FcR) binding affinity, and FcR abundance in syncytiotrophoblasts and endothelial cells contribute to inter-subclass competition and potentially inter- and intra-patient antibody transfer heterogeneity. We developed an *in silico* prenatal vaccine testbed by combining a computational model of maternal vaccination with this placental transfer model using the tetanus, diphtheria, and acellular pertussis (Tdap) vaccine as a case study. Model simulations unveiled precision prenatal immunization opportunities that account for a patient's anticipated gestational length, placental size, and FcR expression by modulating vaccine timing, dosage, and adjuvant. This computational approach provides new perspectives on the dynamics of maternal-fetal antibody transfer in humans and potential avenues to optimize prenatal vaccinations that promote neonatal immunity.

## Author summary

Newborns are vulnerable to infections due to their naïve immune system. Maternal antibodies transferred through the placenta protect the newborn while their own immune system acclimates to the environment outside of the womb. As the dampened immune response in early life presents a challenge to newborn vaccination, maternal vaccines are used to boost pathogen-specific antibody transfer. Despite the exciting therapeutic

equations and in vitro Transwell assay data are available as Supplemental files.

**Funding:** Research reported in this publication was supported by the Thomas F. and Kate Miller Jeffress Memorial Trust and Bank of America under Jeffress Trust Award in Interdisciplinary Research (Principal Investigator: SD, 2022-2023) and Jeffress Trust Award Program in Research Advancing Health (Principal Investigator: SD, 2023-2026). These awards provided funding for the research and partial salary support to SD and REW. Research reported in this publication was additionally supported by the National Institute of General Medical Sciences of the National Institutes of Health under Systems & Biomedical Data Science training program (T32-GM145443, awarded one year salary support to REW from 2022-2023). The funders had no role in study design, data collection and analysis, decision to publish, or preparation of the manuscript.

**Competing interests:** The authors have declared that no competing interests exist.

potential of this approach, few maternal vaccines are currently in use and experimental limitations pose a challenge to optimizing maternal vaccine strategies. To uncover mechanistic insights into this process and inform vaccine design, we developed the first computational model of placental antibody transfer. Model simulations revealed antibody abundance and placental Fc receptor expression determine transfer efficiency. We use this computational model to perform *in silico* immunization optimization experiments, revealing two key insights: (1) second trimester vaccination may be an effective population-level strategy for all neonates and (2) vaccination programs can be optimized in a vaccine- and patient-specific manner to maximize transfer of vaccine-induced antibodies. Ultimately, this model will expedite translation of novel immunization strategies from bench to bedside.

## Introduction

Neonates are vulnerable to infections due to their tolerogenic immune phenotype; for the same reason, neonatal vaccinations have been met with limited success to date [1]. To provide passive immunity while the neonatal immune system adapts to the environment *ex utero*, maternal immunoglobulin G (IgG) is selectively transferred across the placenta during gestation. Not only do these maternal antibodies confer neonates with passive early life immunity against infectious diseases, but also they shape the trajectory of neonatal immune development by priming the cellular immune response [2–4]. Despite this crucial role, the molecular underpinnings of IgG transplacental transport are incompletely described to date.

Though the placenta obstructs maternal-fetal transport of most large molecules, maternal IgG is selectively transported by way of Fc receptor (FcR)-mediated transcytosis. To enter fetal circulation, IgG must undergo transcytosis through two key cellular layers of the placenta: syncytiotrophoblasts (STBs) and endothelial cells (ECs). The neonatal Fc receptor (FcRn) is known to mediate IgG transport through STBs, but whether this mechanism is sufficient for IgG transport through ECs remains unclear [5–7]. In addition to FcRn, STBs and ECs express several Fc gamma receptor (FcγR) isoforms with heterogeneous affinity for IgG [8–10]. These classical FcγRs may contribute to observed selective placental "sieving" of IgG dependent upon fragment crystallizable (Fc) characteristics, e.g., IgG subclass and N-linked glycosylation [11–14], yet there is conflicting evidence surrounding the involvement of FcγRs in IgG transport [9,10,15,16]. Recent studies hypothesized a role for STB-bound FcγRIIIa or EC-bound FcγRIIb in IgG transfer [9,12,17–21], yet others showed FcRn is sufficient for IgG transport in isolated placental ECs *in vitro* and in mouse placenta *in vivo* [14,22,23]. Though informative, the experimental methods underlying these findings do not capture the dynamic nature or time scale of placental antibody transport as it occurs in humans. To disentangle the involvement of non-canonical FcγRs in IgG sieving, innovative methods that recapitulate the longitudinal dynamics of this fundamental process are needed.

Placental antibody transfer can be leveraged by maternal prenatal vaccines which boost pathogen-specific IgG transport to the neonate [24]. Identifying improved immunization strategies to maximize neonatal antibody titers is non-trivial. Placental growth and development dynamically regulating IgG transport coupled with known maternal immune adaptations during pregnancy together define a unique immunization design space which poses a challenge to empirical vaccine optimization by clinical trials alone [25]. Predictive kinetic-dynamic modeling can be employed to rationally design vaccines that maximize IgG transfer to the neonate. The ability to make *in silico* predictions of vaccine-induced antibody transfer

would enable pre-clinical dosing strategy studies, expediting translation of novel therapeutics from bench to bedside.

In the United States and other developed countries, expecting mothers are routinely vaccinated against tetanus, diphtheria, and acellular pertussis (Tdap) during the early third trimester [26], but emerging evidence suggests that this recommendation may not optimally protect the entire population. First, it has been shown that maternal Tdap immunization earlier in gestation results in higher pertussis toxin (PT)- and filamentous hemagglutinin (FHA)-specific IgG in infants regardless of gestational length [26–28]. Second, current vaccination efforts are designed to elicit high antibody transfer to term neonates, yet third trimester immunization may not allot sufficient time for the mother to mount a humoral immune response and subsequently transfer antibodies to preterm neonates [28]. Collectively, this evidence supports the potential for personalized vaccine approaches that account for factors such as maternal baseline IgG titer, immunization history, placental function, and risk of preterm delivery to maximize pathogen-specific IgG transfer to the neonate, especially among premature neonates.

To uncover the molecular regulators of placental antibody transport and to inform the development of personalized immunization approaches, we developed the first computational model of human placental IgG transfer. In a case study on Tdap immunization, we use this model as an *in silico* testbed for prenatal vaccine design and identify potential strategies to improve transfer of vaccine-induced antibodies, both at a patient-specific and population level. Ultimately, this model-driven investigation sheds light on the dynamic regulation of maternal-fetal IgG transfer and provides a foundation to develop precision vaccine approaches which promote neonatal immunity.

## Results

### Mechanistic model recapitulates IgG subclass-specific placental transfer

To elucidate mechanisms of IgG transfer and selective "sieving", we devised a dynamic model of IgG transplacental transfer. The model consists of ordinary differential equations (ODEs) describing IgG mass transport through the distinct layers comprising the maternal-fetal interface: STB FcRn-mediated transcytosis, diffusion through intervillous stroma, and EC FcRn- and FcγRIIb-mediated transcytosis (Fig 1A) (see S1 Appendix for model equations). FcRn is widely implicated in placental IgG transport; we additionally chose to incorporate FcγRIIb into the model due to its expression in term placental ECs and demonstrated role in EC transcytosis *in vitro* [9,19,29]. To account for increasing Fc receptor expression across gestation as observed in human and rat placenta, the total bound and unbound Fc receptors were input to the model as time-varying offline variables [30,31] (S1 Fig and S1 Appendix). Model parameters were derived from literature when available, and 11 unknown parameters were estimated by fitting predictions of fetal IgG to data obtained from cordocentesis in human pregnancies (Fig 1B and Table 1) [32]. Parameter estimation was performed using CaliPro software as described elsewhere [32,33] (see Methods). With this optimized parameter set, the model recapitulated dynamics of both bulk and subclass-specific IgG transfer, predicting the IgG subclass transfer hierarchy most frequently reported in literature (IgG1 > IgG3 > IgG4 > IgG2) (Figs 1C and S2) [11–13,32,34–36].

### Transcytosis rate parameters and Fc receptor expression regulate IgG transfer efficiency and selectivity

To determine the contribution of each parameter to the subclass-specific dynamics predicted by the model, we performed global sensitivity analyses. We generated 1000 randomly

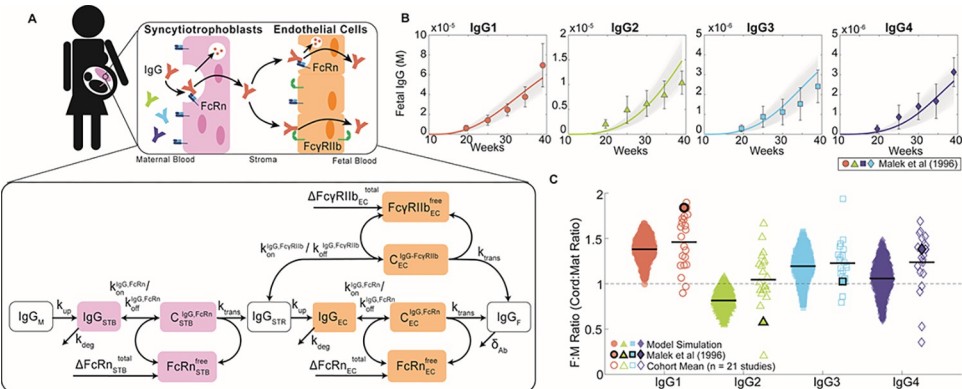

**Fig 1. Mechanistic model captures IgG subclass-specific transfer dynamics.** (A) The schematic depicts the ODE model representing transport of IgG through the placenta, mapped onto a graphical representation of the maternal-fetal interface. Reaction rate constants associated with each process are depicted on the corresponding arrow. Each box represents a molar concentration in each compartment. "C" denotes complexes of bound IgG and Fc receptors either in endosomes (FcRn) or at the cell surface (FcγRIIb). $\Delta FcRn_{STB}^{total}$, $\Delta Fc\gamma RIIb_{EC}^{total}$, and $\Delta FcRn_{STB}^{total}$ represent synthesis of new FcRn and FcγRIIb molecules over time as the placenta grows. For simplicity, the schematic represents bulk IgG transport, whereas the model includes separate equations for each IgG subclass. (B) Simulated fetal IgG subclass levels vs. time (lines) are overlaid with the data used for parameter estimation. The overlaid data are the mean and standard deviation of IgG subclasses in the umbilical cord between 17–41 weeks gestation (n = 107) measured using cordocentesis by Malek *et al* [32]. Gray lines represent 1000 perturbation simulations resulting from sampling parameters from the ranges determined by CaliPro. (C) The swarm charts show the fetal/maternal ratio (F:M ratio) for each IgG subclass at 40 weeks gestation resulting from 1000 simulations sampling from the optimized range of parameters (filled symbols) compared to the mean cord/maternal ratio for each IgG subclass compiled across cohort studies mined from the literature (hollow symbols). Each symbol represents a single simulation or cohort average. Sample means are indicated by black lines. The symbols outlined in black represent the mean of IgG subclass cord/maternal transfer ratios at parturition from the human study used during parameter optimization [32]. Dashed line at F:M ratio = 1 indicates equal concentration in maternal and fetal blood at parturition.

perturbed parameter sets using Latin Hypercube Sampling (LHS) from uniform distributions spanning the maximum likelihood parameter ranges determined by CaliPro [33,40,41]. For each parameter set, the predicted fetal IgG concentration at 40 weeks was calculated. To determine the parameters with the greatest influence on total IgG transfer (Fig 2A and 2B) and on differential transfer between IgG subclasses defined as entropy of subclasses (Fig 2C and 2D), orthogonalized partial least squares regression (OPLSR) models were developed using the 1000 perturbed parameter sets to predict either outcome (Methods) [42]. Variable importance in projection (VIP) scores revealed that total IgG transfer was most sensitive to the rate of IgG transcytosis by STBs and ECs ($k_{trans}$), FcRn expressed by STBs, and FcγRIIb expressed by ECs (Fig 2B). On the other hand, all volume parameters negatively correlated with IgG transfer as increasing the volume of either compartment dilutes the IgG concentration. To reveal parameters that contribute to the difference in IgG transfer among subclasses, we built a second OPLSR model predicting the entropy among IgG subclasses (Methods) (Fig 2C and 2D). Interestingly, VIP score analysis showed FcγRIIb expressed by ECs and $k_{trans}$ contribute most to the distinct transfer rates among IgG subclasses. FcγRIIb is a low-affinity IgG receptor with highest affinity for subclasses IgG3 and IgG4, suggesting that it constitutes a rate-limiting factor of IgG transfer which fine-tunes the subclass transfer hierarchy [11,15,29]. Likewise, $k_{trans}$ acts in an FcR-dependent manner to further amplify the effects of Fc-mediated selectivity. Thus, the rate of cargo internalization and fetoplacental size drives the concentration of bulk IgG in the fetus, whereas FcRn and FcγRIIb collaborate to fine-tune patterns of IgG subclass-specific transfer.

**Table 1. List of optimized model parameters.**

| Parameter | Value | Units | Interpretation | Source |
|---|---|---|---|---|
| $k_{up}$ | 0.0872–0.0982 | L/week | Uptake rate into cells | Optimized |
| $k_{trans}$ | 0.0569–0.0743 | L/week | Transcytosis rate | Optimized |
| $k_{deg}$ | 8.4733–9.0809 | L/week | Lysosomal degradation rate | Optimized |
| $FcRn_{STB}^{total,end}$ | $3.33$–$4.94 \times 10^{-5}$ | M | STB FcRn expression at term | Optimized |
| $Fc\gamma RIIb_{EC}^{total,end}$ | $2.73$–$3.33 \times 10^{-5}$ | M | EC FcγRIIb expression at term | Optimized |
| $FcRn_{EC}^{total,end}$ | $2.36$–$2.86 \times 10^{-6}$ | M | EC FcRn expression at term | Optimized |
| $IgG1_0$ | $3.78 \times 10^{-5}$ | M | Maternal IgG1 concentration | [32] |
| $IgG2_0$ | $1.81 \times 10^{-5}$ | M | Maternal IgG2 concentration | [32] |
| $IgG3_0$ | $2.35 \times 10^{-6}$ | M | Maternal IgG3 concentration | [32] |
| $IgG4_0$ | $2.27 \times 10^{-6}$ | M | Maternal IgG4 concentration | [32] |
| $K_D$, FcRn-IgG1 ($k_{off}/k_{on}$) | $1.25 \times 10^{-8}$ ($0.15/1.2 \times 10^{7}$) | M | FcRn and IgG1 dissociation constant | [37,38] |
| $K_D$, FcRn-IgG2 ($k_{off}/k_{on}$) | $2 \times 10^{-8}$ ($0.15/7.5 \times 10^{6}$) | M | FcRn and IgG2 dissociation constant | [37,38] |
| $K_D$, FcRn-IgG3 ($k_{off}/k_{on}$) | $3.3 \times 10^{-8}$ ($0.15/4.5 \times 10^{6}$) | M | FcRn and IgG3 dissociation constant | [37,38] |
| $K_D$, FcRn-IgG4 ($k_{off}/k_{on}$) | $5 \times 10^{-8}$ ($0.15/3 \times 10^{6}$) | M | FcRn and IgG4 dissociation constant | [37,38] |
| $K_D$, FcγRIIb-IgG1 ($k_{off}/k_{on}$) | $1 \times 10^{-5}$ ($0.15/1.5 \times 10^{4}$) | M | FcγRIIb and IgG1 dissociation constant | [37,38] |
| $K_D$, FcγRIIb-IgG2 ($k_{off}/k_{on}$) | $5 \times 10^{-5}$ ($0.15/3 \times 10^{3}$) | M | FcγRIIb and IgG2 dissociation constant | [37,38] |
| $K_D$, FcγRIIb-IgG3 ($k_{off}/k_{on}$) | $5 \times 10^{-6}$ ($0.15/3 \times 10^{4}$) | M | FcγRIIb and IgG3 dissociation constant | [37,38] |
| $K_D$, FcγRIIb-IgG4 ($k_{off}/k_{on}$) | $5 \times 10^{-6}$ ($0.15/3 \times 10^{4}$) | M | FcγRIIb and IgG4 dissociation constant | [37,38] |
| $V_M$ | 5.552–5.619 | L | Maternal blood volume | Optimized |
| $V_{STB}$ | 0.224–0.258 | L | Total STB endosomal volume | Optimized |
| $V_{STR}$ | 0.165–0.179 | L | Stromal volume | Optimized |
| $V_{EC}$ | 0.247–0.275 | L | Total EC endosomal volume | Optimized |
| $V_F$ | 0.254–0.271 | L | Fetal blood volume | Optimized |
| $\delta_{Ab}$ | 0.02 | L/week | IgG decay rate in the fetus | [39] |

## Endothelial cell-bound FcγRIIb is a key driver of subclass-specific IgG transfer

It is well-established that placental IgG transport increases during the third trimester in humans, so we asked whether increasing FcγRIIb or FcRn expression in ECs drives increased transport efficiency. To gain insight into the expression dynamics of the genes coding for FcγRIIb and FcRn, namely *FCGR2B* and *FCGRT* (FcRn heavy chain), in fetal ECs across gestation, we analyzed two single cell RNA sequencing (scRNA-seq) datasets from published literature spanning three gestational time points: (1) first trimester samples from elective abortions (< 16 weeks), (2) early third trimester samples from preeclamptic cesarean sections (30 weeks), and (3) late third trimester samples from uncomplicated cesarean sections (>37 weeks weeks) [43,44]. Interestingly, *FCGR2B* expression was significantly higher in late compared to early third trimester ECs, but we did not detect a significant difference in *FCGRT* expression between early and late third trimester ECs (Fig 3A and 3B). Conversely, *FCGRT* but not *FCGR2B* transcripts were detected in first trimester ECs, and the ratio of *FCGR2B* to *FCGRT* expression per cell increased across time points (Figs 3C and S3), suggesting that FcγRIIb dynamically regulates IgG transcytosis in ECs.

Previous studies showed that FcRn and FcγRIIb are co-expressed by placental ECs at the protein level [9,22]. Since FcRn has a greater affinity for all IgG subclasses (Table 1), it is unclear whether FcRn alone is responsible for IgG transfer across fetal ECs [22]. To assess the contribution of FcγRIIb to IgG transport, we tested whether FcγRIIb is a determinant of subclass-specific transfer. In the model with FcRn and FcγRIIb co-expressed ($EC_{FcRn,Fc\gamma RIIb}$), we

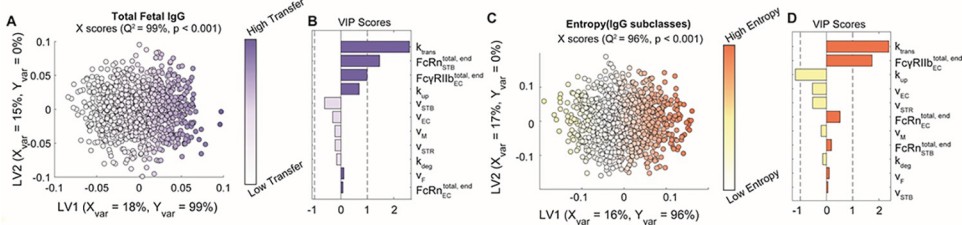

**Fig 2. Key parameters associated with transcytosis mechanism and Fc receptor expression regulate IgG transfer.**
(A-B) An OPLSR model was built to determine parameters predictive of total fetal IgG. (A) The scores plot shows X scores on latent variables 1 and 2 (LV1 and LV2) of simulated fetal IgG concentration at parturition for 1000 randomly perturbed parameter sets. Each circle represents a single perturbation experiment. The points are colored proportional to the resulting simulated concentrations of IgG in the fetus at delivery. The model was orthogonalized such that the direction of maximum variance is in the direction of LV1, capturing 99% of the Y variation. The model performed with high predictability power ($Q^2$ = 99%) and a mean squared error which outperformed 1000 random models with permutated labels ($p < 0.001$). (B) The bar graph depicts the variable importance in projection (VIP) scores, which shows the relative contribution of each parameter to total fetal IgG for a term delivery. VIP scores are artificially oriented in the direction of their loadings on LV1. VIP score > 1 indicates greater-than-average contribution of that parameter. (C-D) An OPLSR model was built to predict entropy among fetal IgG subclass levels at the end of gestation. (C) The scores plot showing X scores on LV1 and LV2 of simulated fetal IgG concentration at parturition for 1000 perturbation parameter sets. Each circle represents a single perturbation experiment. The points are colored proportional to the resulting simulated concentrations of IgG in the fetus at delivery. The model was orthogonalized such that LV1 captures features associated with the highest entropy, accounting for 96% of the Y variation. The model performed with high predictability power ($Q^2$ = 96%) and a mean squared error which outperformed 1000 models based on randomly permutated labels ($p < 0.001$). (D) The VIP scores bar plot shows the relative contribution of each parameter to the entropy of IgG subclasses in fetal blood for a term delivery. VIP scores are artificially oriented in the direction of their loadings on LV1.

found that the predicted subclass transfer hierarchy was IgG1 > IgG3 > IgG4 > IgG2 when FcRn expression was less than 25% of FcγRIIb (Fig 3D), demonstrating that higher relative expression of FcγRIIb to FcRn by ECs drives subclass-specific IgG placental transfer. As FcγRIIb binds subclasses IgG1, IgG3, and IgG4 with greater affinity than IgG2, we hypothesized that FcγRIIb is necessary to predict preferential transfer of IgG1, IgG3, and IgG4 over IgG2. To test this hypothesis, we compared model-predicted fetal-maternal IgG ratios from simulations assuming ECs solely express FcRn ($EC_{FcRn}$) or FcγRIIb ($EC_{Fc\gamma RIIb}$). The $EC_{FcRn}$ model did not capture the preferential transfer of IgG1, IgG3, and IgG4 at any FcRn expression

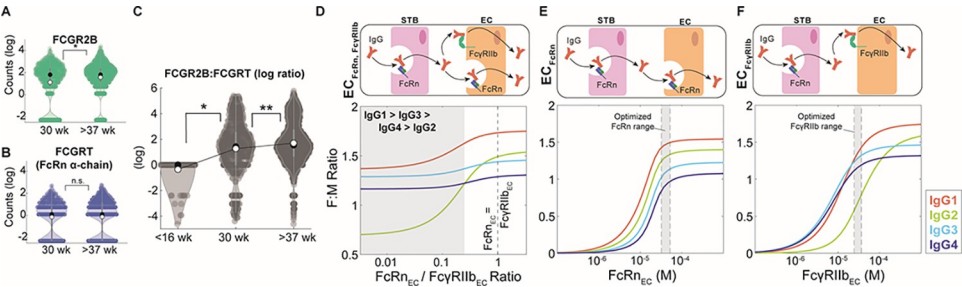

**Fig 3. Endothelial cell-bound FcγRIIb is a key driver of subclass-specific IgG transfer.** (A-B) Violin plots show *FCGR2B* (A) and *FCGRT* (B) transcript levels in preterm, preeclamptic ECs (30 weeks) compared to term, healthy ECs (>37 weeks) from single cell RNA sequencing data. Black circles represent the sample median, white circles represent the sample mean. (C) Violin plots show the log-transformed ratio of *FCGR2B* to *FCGRT* counts in ECs from single cell RNA sequencing data from two separate cohorts spanning 3 gestational time points: <16 weeks, 30 weeks, and >37 weeks. Black circles represent the sample median, white circles represent the sample mean. (D-F) Model-simulated IgG subclass transfer ratios as a function of EC Fc receptor expression with ECs expressing FcRn and FcγRIIb (D, $EC_{Fc\gamma RIIb-FcRn}$), only FcRn (E, $EC_{FcRn}$), or only FcγRIIb (F, $EC_{Fc\gamma RIIb}$). The corresponding schematics depict the simulation conditions. Statistical significance in (A-C) was determined by a Wilcoxon rank sum test (* $p < 0.01$, ** $p < 0.001$, n.s. not significant).

level (Fig 3E). Interestingly, in the $EC_{Fc\gamma RIIb}$ model the subclass transfer hierarchy shifted from IgG3 > IgG4 > IgG1 > IgG2 to IgG1 > IgG2 > IgG3 > IgG4 with increasing FcγRIIb expression, indicating FcγRIIb drives the observed subclass transfer hierarchy in humans (Fig 3F). Together, these data implicate FcγRIIb in ECs as a key regulator of antibody transfer dynamics across gestation.

## IgG subclasses compete for receptor-mediated transcytosis

Our finding that FcγRIIb limitation plays a key role in determining the subclass transfer hierarchy suggests that maternal IgG subclasses compete for receptor-mediated transcytosis. On the other hand, variation in IgG subclass abundances across individuals may contribute to patient-specific differences in IgG subclass transfer. In agreement with this hypothesis, the placental transport model predicted that all four IgG subclasses transferred less efficiently when mixed compared to each subclass in isolation of the others and the effect of mixing was greater for subclasses with low abundances in maternal serum at low FcγRIIb expression levels (S4 Fig). To further elucidate the role of inter-subclass competition in determining transport efficiency, we explored the effect of IgG1 on IgG4 transcytosis in ECs in a separate model of transcytosis across ECs only (see Methods). We chose to model IgG1 and IgG4 as a case study because they have the highest and lowest abundance in maternal serum, respectively, yet both generally exhibit a cord/maternal ratio > 1 [11,32]. Based on a simple kinetic reaction model, the inter-subclass competition can be described by the following closed-form equation:

$$C_{IgG4-FcRn} = \frac{FcRn_0[IgG4]}{K_{D,IgG4}\left(1 + \frac{[IgG4]}{K_{D,IgG4}} + \frac{[IgG1]}{K_{D,IgG1}}\right)}$$

Where $C_{IgG4-FcRn}$ is the concentration of bound IgG4 and FcRn, $FcRn_0$ is the total concentration of receptor, [IgG1] and [IgG4] are concentrations of maternal IgG1 and IgG4, and $K_{D,IgG1}$ and $K_{D,IgG4}$ are their respective equilibrium dissociation constants (see derivation in Methods). Intuitively, $C_{IgG4-FcRn}$ is a function of IgG1 abundance and affinity for FcRn, such that either increasing [IgG1] or decreasing $K_{D,IgG1}$ (i.e., increasing IgG1-FcRn affinity) decreases $C_{IgG4-FcRn}$ and consequently, IgG4 transcytosis.

To validate and further characterize competition among IgG subclasses, we used an *in vitro* system to model IgG transcytosis through human umbilical vein endothelial cells (HUVEC) cultured on Transwell permeable membranes (Fig 4A) (Methods). In parallel, we devised a distinct, simplified mechanistic model of receptor-mediated transcytosis in ECs and optimized its parameters by fitting to temporal data of IgG4 transcytosis in the HUVEC experimental system ([IgG4apical] = 0.5 or 0.125 mg/ml) (S5 Fig and Table 2). To recapitulate the physiological conditions where FcR is limited, we varied IgG4 levels to determine the concentration of IgG4 where IgG4 transfer no longer increases with increasing apical IgG4 levels. Although HUVEC expresses FcRn but not FcγRIIb, our findings remain valid because the affinities of IgG1 and IgG4 for either receptor are on the same order of magnitude and thus the dynamics of subclass competition are comparable [37]. Our mechanistic model predicted that IgG4 transcytosis saturates at [IgG4apical] > 0.33 mg/ml, indicating FcRn limitation; we validated this finding in an analogous *in vitro* experiment in the Transwell system (Fig 4B). To quantify the effect of IgG1 on IgG4 transcytosis, we simulated the effect of increasing IgG1 concentration when IgG4 was fixed at a concentration just below the FcRn saturation point ([IgG4apical] = 0.2 mg/ml). Our model predicted that IgG4 transcytosis would not be affected by the presence of IgG1 until the total IgG concentration exceeded the saturation point—that is, [IgG1apical] + [IgG4apical] > 0.33 mg/ml. In agreement with our computational model prediction, increasing IgG1 caused IgG4 transcytosis to decrease only when the total IgG concentration exceeded the FcRn

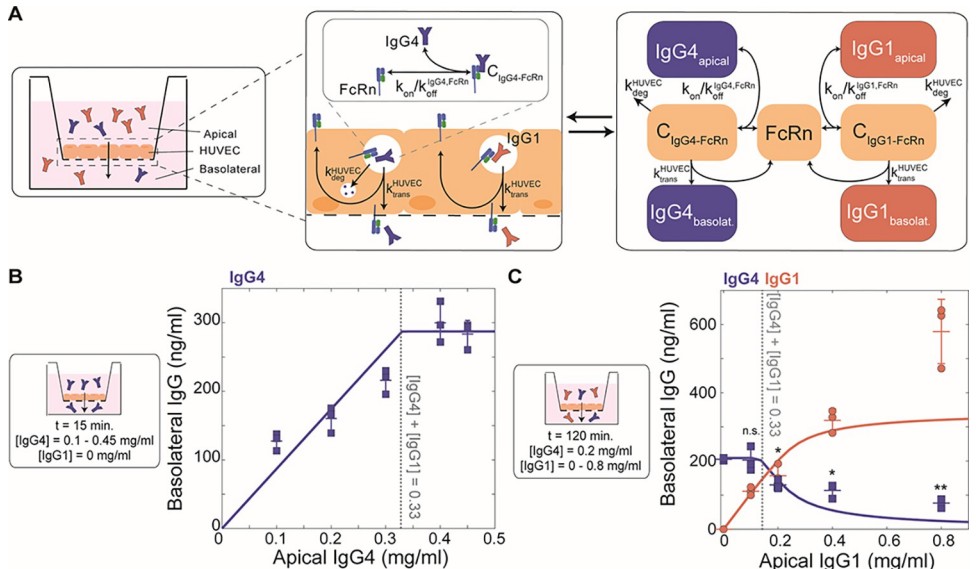

**Fig 4. IgG1 competes with IgG4 for FcRn-mediated transcytosis in HUVECs.** (A) Schematic showing a cross-sectional view of the Transwell system and corresponding mechanistic model representation of FcRn-mediated IgG4 and IgG1 transport through HUVEC. Left panel: Arrow indicates the direction of transfer (apical to basolateral). Center and right panels: Reaction rate constant associated with each process is depicted on the corresponding arrow. (B) Model-predicted IgG4 transcytosis (y-axis) with increasing apical concentration (x-axis) (solid line) is plotted against experimental data points (mean and standard deviation from three independent biological replicates). The corresponding experimental conditions are displayed in the panel on the left. The dashed line represents the model-predicted and experimentally validated transcytosis saturation point ($[IgG_{total}] = [IgG4] = 0.33$ mg/ml). (C) Model-predicted IgG4 and IgG1 transcytosis (y-axis) with increasing apical concentration of IgG1 (x-axis) (solid lines) is plotted against experimental data points (mean and standard deviation from three independent biological replicates). The corresponding experimental conditions are displayed in the panel on the left. The dashed line represents the model-predicted and experimentally validated transcytosis saturation point ($[IgG_{total}] = [IgG4] + [IgG1] = 0.33$ mg/ml. Statistical comparisons shown are between IgG4 in the absence of IgG1 (far left condition) and all combinations of IgG1 and IgG4 mixed using a one-tailed two-sample t-test (* $p < 0.05$, ** $p < 0.001$, n.s. not significant).

saturation point (Fig 4C). IgG subclass-specific transfer is thus a function of subclass abundances and FcR affinity, and competition for FcR-mediated transcytosis occurs when FcR expression is limited.

## Optimal Tdap vaccine strategies depend on gestational age, kinetics of the maternal IgG response to Tdap, and Fc receptor expression

Considering the new insights into the dynamic regulation of antibody transfer by FcRs revealed by our model, we sought to uncover key features influencing the transfer of vaccine-induced antibodies which could be considered in precision immunization approaches. To this end, we developed an *in silico* immunization testbed by layering simulated kinetics of the

**Table 2. Optimized HUVEC transcytosis model parameters.**

| Parameter | Value | Units | Interpretation | Source |
|---|---|---|---|---|
| FcRn | 2200 | nM | HUVEC FcRn expression level | Optimized |
| $k_{deg}^{HUVEC}$ | 0.01 | 1/minute | Lysosomal degradation rate | Optimized |
| $k_{trans}^{HUVEC}$ | $2.5 \times 10^{-5}$ | 1/minute | Transcytosis rate | Optimized |
| $K_D$, FcRn-IgG1 ($k_{off}/k_{on}$) | $1.25 \times 10^{-8}$ ($0.15/1.2 \times 10^{7}$) | M | FcRn and IgG1 dissociation constant | [37,38] |
| $K_D$, FcRn-IgG4 ($k_{off}/k_{on}$) | $5 \times 10^{-8}$ ($0.15/3 \times 10^{6}$) | M | FcRn and IgG4 dissociation constant | [37,38] |

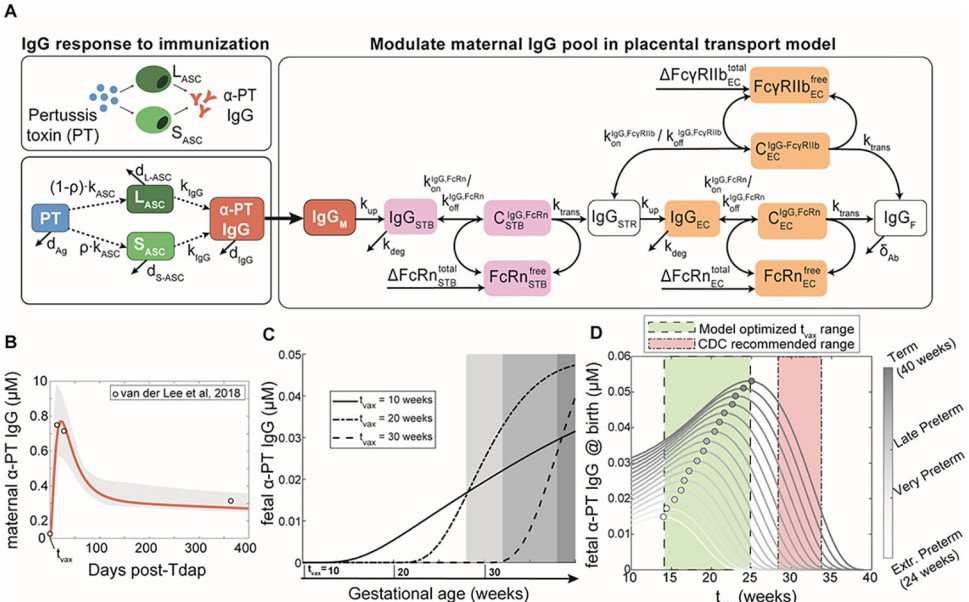

**Fig 5. Optimal Tdap immunization time is gestational age dependent.** (A) Schematic shows mechanistic model of maternal plasma cell antibody secretion post-vaccination. Vaccine-induced maternal IgG kinetics are integrated into the maternal compartment of the placental IgG transport model, shown in the righthand box. Dashed arrows indicate regulatory processes, solid lines represent processes modeled using mass action kinetics. $L_{ASC}$: Long-lived antibody secreting cell, $S_{ASC}$: short-lived antibody-secreting cell, Ag: antigen. (B) Simulated maternal anti-pertussis IgG concentrations vs. time is plotted with a solid line and overlaid with measured IgG concentrations following Tdap vaccination in women of childbearing age from published literature [46] (n = 105). The 95% confidence intervals are shown with light gray shading. (C) Simulated fetal anti-pertussis toxin IgG concentrations are plotted against time following maternal vaccination at three sample vaccination times (indicated along the x-axis): $t_{vax}$ = 10, 20, and 30 weeks gestational age. Shaded regions correspond to gestational length groups (left to right): extremely preterm, very preterm, late preterm, term. (D) Simulated anti-pertussis toxin IgG in the fetus at the time of delivery (y-axis) is plotted over a range of simulated vaccination times (x-axis) and gestational ages (color intensity, color bar on right). The optimal vaccination time and corresponding fetal IgG levels are marked with a circle. The current CDC-recommended maternal vaccination window is highlighted in red, and the model-predicted optimal vaccination range encompassing all gestational age groups is highlighted in green.

maternal antibody response post-immunization into the placental antibody transfer model (Fig 5A). As a case study of a maternal vaccine currently in use, we adapted an existing model of plasma cell activation and antibody secretion following malaria infection to recapitulate IgG responses to Tdap immunization in women of child-bearing age (Methods) (Fig 5B and Table 3) [45,46]. A dose of antigen representing maternal immunization was applied to the system at three representative gestational ages ($t_{vax}$) of 10, 20, and 30 weeks to stimulate antibody secreting cell proliferation and anti-pertussis toxin IgG (α-PT IgG) secretion and transfer to the fetus (Fig 5C). Interestingly, the interplay of maternal IgG response kinetics and placental transfer results in different IgG transfer dynamics depending on $t_{vax}$. This non-intuitive, gestational-age dependent effect on transfer dynamics emphasizes the importance of *in silico* predictions to determine optimal immunization schedules.

We next performed simulations over a range of possible immunization times to determine the optimal $t_{vax}$ for newborns born at different weeks of gestation (Fig 5D). Though the correlate of protection for pertussis infection still remains elusive, some possible roles of IgG in pertussis infection include neutralization and opsonophagocytosis [48, 49]. For the purposes of this study, we assumed that neutralizing antibodies confer protection against pertussis toxin in the neonate and therefore the optimal $t_{vax}$ is defined as the gestational age at immunization resulting in the highest α-PT IgG titer in the fetus at birth. Empirical studies have previously

**Table 3. Pertussis immunization model parameters.**

| Parameter | Value | Units | Interpretation | Source |
|-----------|-------|-------|----------------|--------|
| $k_{ASC}$ | 0.6 | 1 / $10^6$ PBMCs / day / Ag | Rate of plasma cell generation | Optimized |
| $k_{IgG}$ | 0.0475 | mg / ml * ($10^6$ PBMCs) / days | Rate of antibody production by plasma cells | Optimized |
| $Ag_0$ | 100 | mg/ml | Initial dose of antigen | Optimized |
| $\rho$ | 96 | % | Proportion of short-lived ASCs ($S_{ASC}$) | [45] |
| $\delta_{Ag}$ | 0.15 | 1 / day | Antigen decay rate | [47] |
| $\delta_{IgG}$ | 0.033 | 1 / day | Antibody decay rate | [45] |
| $\delta_{S-ASC}$ | 0.173 | 1 / day | Short-lived antibody-secreting cell ($S_{ASC}$) decay rate | [45] |
| $\delta_{L-ASC}$ | $6.6 \times 10^{-4}$ | 1 / day | Long-lived antibody secreting cell ($L_{ASC}$) decay rate | [45] |

shown that earlier vaccination may be an effective strategy to protect preterm neonates who typically receive a lower quantity of maternal antibodies [26–28]. Using our immunization testbed, we first asked how the optimal Tdap administration schedule varies for the average patient delivering at different gestational time points (i.e., term, late preterm, very preterm, or extremely preterm). In the case of term gestation, the model-predicted optimal $t_{vax}$ was during the 25[th] week of gestation, but Tdap immunization as early as the 10[th] week still resulted in greater than half of the optimal IgG concentration being transferred to term newborns (Fig 5D). Optimal $t_{vax}$ scaled with increasing gestational length, suggesting that risk for pre-term birth should be considered when determining prenatal immunization schedules (Fig 5D). Interestingly, we found the optimal $t_{vax}$ fell during the second trimester (between 14–25 weeks) regardless of preterm birth status, which is earlier than the window recommended by the Center for Disease Control and Prevention (CDC) (28–34 weeks) and in agreement with studies reporting higher titer of pertussis antibodies in the fetus following maternal vaccination during second trimester as opposed to third trimester [26, 28, 50]. This optimized Tdap vaccination window was unaffected by the magnitude of the maternal α-PT IgG response but did depend on the existence of a transient spike in maternal α-PT IgG driven by the percentage of $S_{ASC}$ ($\rho$) (S6 and S7 Figs). While α-PT IgG responses following Tdap are generally character-ized by a short-lived spike in IgG, variation in the IgG response kinetics for other target anti-gens should be considered when applying this model to other vaccines. Thus, the optimal Tdap immunization schedule is gestational length-dependent, but a revised vaccination win-dow earlier in gestation could be an effective population-level strategy to ensure coverage for preterm and term newborns alike.

As other sources of inter-patient variability besides gestational length may influence the effectiveness of prenatal vaccination, we sought to identify further opportunities for patient-specific immunization approaches. To determine key variables that should be considered in immunization schedules, we performed global sensitivity analysis to uncover parameters with the greatest influence on optimal $t_{vax}$. We optimized $t_{vax}$ in simulations from 1000 perturbed parameter sets, then built an OPLSR model to predict the optimal $t_{vax}$ given each parameter set (Fig 6A and 6B, Methods). Parameters strongly associated with earlier optimal immunization times included $FcRn_{STB}^{total,end}$ and $v_F$; $FcRn_{EC}^{total,end}$ and $Fc\gamma RIIb_{EC}^{total,end}$ similarly had negative loadings on latent variable 1 (LV1), as did several other volume parameters. Intuitively, higher compart-mental volumes decrease transfer of vaccine-induced antibodies because IgG concentration is diluted in high volume compartments, and thus earlier vaccination enables more IgG to accu-mulate in the fetus to compensate for this dilution. Likewise, higher $FcRn_{STB}^{total,end}$ facilitates IgG transport into the placental stroma making IgG subsequently available for trans-EC transport. If sufficient $FcRn_{STB}^{total,end}$ is expressed, earlier immunization corresponds to increased levels of

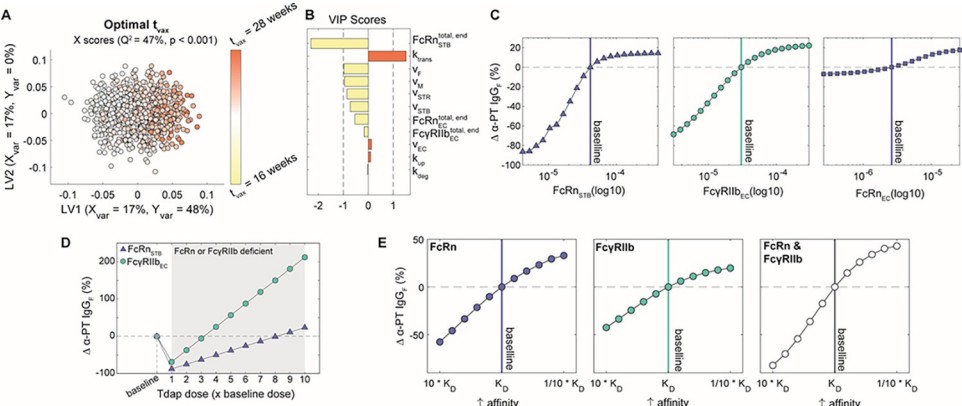

**Fig 6. Model reveals opportunities for personalized vaccination strategies.** (A-B) The OPLSR model predicts the optimal immunization time ($t_{vax}$, weeks gestational age) from perturbed parameter sets. (A) The scores plot depicts the X scores on LV1 and LV2 of the optimal $t_{vax}$ for 1000 perturbed parameter sets. Each circle represents a single perturbation experiment colored proportionally to the predicted optimal $t_{vax}$. The model was orthogonalized such that LV1 is in the direction of maximum variance. (B) The bar plot shows VIP scores that quantify the relative contribution of each parameter to the optimal $t_{vax}$. VIP scores are artificially oriented in the direction of their loadings on LV1. The model performed with predictive power ($Q^2 = 47\%$) and with a mean squared error lower than 1000 models based on randomly permutated labels ($p < 0.001$). (C) The value of $FcRn_{STB}^{total,end}$ (left), $Fc\gamma RIIb_{EC}^{total,end}$ (center), or $FcRn_{EC}^{total,end}$ (right) was varied from 10-fold lower to 10 times higher than the mean of their optimal range determined by CaliPro. Vaccination was simulated at the optimal time for a term newborn ($t_{vax} = 25$ weeks) for each FcR condition and the resulting percent changes in anti-pertussis toxin IgG ($\alpha$-PT IgG) in a newborn delivered at 40 weeks is depicted. The dashed line represents the baseline simulation of an average patient using the optimized values of $FcRn_{STB}^{total,end}$, $Fc\gamma RIIb_{EC}^{total,end}$, or $FcRn_{EC}^{total,end}$, which are marked by vertical lines. (D) The chart shows the effect of increasing vaccine dosage on $\alpha$-PT IgG transfer for patients with a 10-fold deficiency in $FcRn_{STB}^{total,end}$ or $Fc\gamma RIIb_{EC}^{total,end}$ expression. Dosages are shown as fold-changes from the baseline simulation dosage. (E) The effect of increasing vaccine-induced $\alpha$-PT IgG affinity for FcRn, Fc$\gamma$RIIb, or FcRn and Fc$\gamma$RIIb on $\alpha$-PT IgG transferred to a fetus delivered at term.

IgG accumulated in the stroma, readily available for trans-EC transport. On the other hand, $k_{trans}$ corresponds to later optimal immunization timing because if transcytosis efficiency increases, it confers greater benefit to timing the maternal IgG spike post-vaccination when FcRn and Fc$\gamma$RIIb are expressed in higher quantities. Therefore, differences in fetoplacental size and function across the population may play a role in determining the ideal patient-specific immunization program.

With the assumption that maternal IgG protects the fetus via neutralization, the effectiveness of maternal vaccination is ultimately determined by the concentration of maternal antigen-specific IgG in the neonate. Having observed that FcRn and Fc$\gamma$RIIb are key predictors of IgG transport efficiency (Fig 2A), we next investigated the relationship between FcR expression and $\alpha$-PT IgG transfer efficiency. Consistent with our previous observation that IgG transfer efficiency is highly sensitive to $FcRn_{STB}^{total,end}$ and $Fc\gamma RIIb_{EC}^{total,end}$, we observed that a 10-fold decrease in $FcRn_{STB}^{total,end}$ or $Fc\gamma RIIb_{EC}^{total,end}$ diminished $\alpha$-PT IgG transfer by 86 or 69% respectively, whereas $FcRn_{EC}^{total,end}$ had a minor impact on $\alpha$-PT IgG in the fetus (Fig 6C). Given that FcR expression is likely variable across the population, we sought to determine vaccination modifications that could potentially compensate for deficiency in either receptor. First, we found that patients with a 10-fold decrease in $FcRn_{STB}^{total,end}$ could transfer comparable levels of $\alpha$-PT IgG to their fetus by increasing the vaccine dosage 8-fold, and a 10-fold decrease in $Fc\gamma RIIb_{EC}^{total,end}$ could be compensated for by increasing the dosage 3-fold (Fig 6D). Naturally, vaccine dosage can only be increased to the extent that it does not induce toxicity. Rather than increasing the *quantity* of maternal vaccine-induced IgG, we asked whether modulating the

*affinity* of vaccine-induced IgG to FcRs (*e.g.* via Fc glycosylation modulation) could improve placental transfer. While precision vaccination is a relatively new concept, recent studies have demonstrated that adjuvants play a key role in driving the glycan profile of vaccine-induced IgG [51, 52]. Coincidentally, it has been shown that decreasing core fucosylation can improve IgG binding to FcγRIIb and terminal galactosylation increases IgG affinity to FcRn [12, 53, 54]. Our model predicted that decreasing $K_D^{IgG1-4,FcRn}$ by 10-fold increased α-PT IgG transfer by 30%, and a similar relationship was observed for $K_D^{IgG1-4,Fc\gamma RIIb}$ (Fig 6E). Interestingly, the effect of increasing affinity for both receptors was greater than the sum of its parts, suggesting affinity for both receptors can be targeted simultaneously to improve prenatal vaccine efficacy.

## Discussion

Placental antibody transfer is key to protect neonates against infection and to shape the trajectory of early-life immune development [55]. Recently, maternal prenatal immunization has been employed to boost antigen-specific antibody transfer to the fetus, providing targeted immunity to the newborn [56]. Current prenatal vaccine regimens are applied at the population level, but variation in gestational length, placental size and function, and maternal immune responses present opportunities for patient-specific vaccine approaches. To this end, we developed a computational model to elucidate the features regulating maternal-fetal IgG transport and to inform prenatal immunization strategies. Our modeling approach enabled longitudinal probing of placental antibody transfer dynamics, shedding light on an otherwise black-boxed system. In contrast to prior studies attempting to uncover mechanisms of selective antibody transfer through a variety of *in vitro*, *ex vivo*, and *in vivo* approaches, our computational approach recapitulates the dynamics of this process as it occurs in humans [14,21,57,58].

Our dynamic model predicted subclass-specific IgG transfer ratios that agree with the trend most commonly reported in the literature: IgG1 > IgG3 > IgG4 > IgG2 [11]. While this phenomenon of subclass selectivity has long been recognized, our model provides a mechanistic basis for selective transfer of IgG subclasses based on competition for binding to FcRs that are expressed in limited quantities. Previous studies have demonstrated that competition for FcRn binding contributes to the short half-life of IgG3 [57]. Our study builds upon this finding by revealing that FcγRIIb in placental ECs similarly contributes to selective transfer by preferentially binding IgG1, IgG3, and IgG4 with the highest affinity. Moreover, we built a mathematical model that quantifies competition between two IgG subclasses as a function of FcR expression level, maternal IgG subclass abundance, and Fc-FcR interaction affinity. Thus, the subclass transfer hierarchy of IgG1 > IgG3 > IgG4 > IgG2 can be explained by each subclass' relative abundance and affinity for FcRn and FcγRIIb. Not only that, but inter-patient variability in subclass abundances, IgG-FcR affinity arising from differential IgG glycosylation, and placental FcR expression can explain discrepancies in this subclass transfer hierarchy across individuals and patient cohorts.

Previous studies have identified independent mechanisms of both FcγRIIb- and FcRn-mediated EC transcytosis in placental ECs [22,29], yet the role of FcγRs in placental transport remains highly contested [14,15,21]. Our computational framework enabled us to disentangle the contributions of FcRn and FcγRIIb to EC transcytosis, revealing that in an average placenta FcγRIIb drives selectivity of IgG transfer and may play a key role in dynamic regulation of this process across gestation. Together, these observations support the potential role of non-canonical FcγRs driving antibody transfer dynamics and provide a mechanistic basis for previously reported placental IgG "sieving" [12,13,59]. Further evidence implicating FcγRIIb in placental transfer include that (1) our analysis of scRNA-seq data revealed an increase in *FCGR2B*

transcript counts across gestation and (2) placental ECs express the splice variant FcγRIIb2—the isoform capable of endocytosis and expressed by liver sinusoidal endothelial cells, B cells, and macrophages [17,60,61]. In the same vein, previous studies indicate a potential role for FcγRIIIa in STB transcytosis [12,15,35]. However, the data supporting FcγRIIIa involvement in placental transfer are purely associative and lack the necessary mechanistic insight to substantiate the inclusion of FcγRIIIa in this model [12,17]. To disentangle the roles of FcRn, FcγRIIb, and FcγRIIIa in IgG transcytosis, future studies that uncover the expression patterns and dynamics of these receptors in human placenta and/or experimentally test their effect on IgG transcytosis are warranted.

This unique modeling approach allowed us to predict optimized Tdap immunization schedules and identify features affecting optimal Tdap timing during pregnancy, with broad implications for prenatal vaccination both at the population level and as a basis for patient-specific vaccine approaches. First, the model predicted that the optimal Tdap immunization time for an average patient to maximize α-PT IgG transfer is during the second trimester for term and preterm neonates alike, aligning with key findings from other recent studies that Tdap immunization earlier in gestation provides higher α-PT and α-FHA IgG in the newborn [28]. While this optimal Tdap administration window was not sensitive to the magnitude of the maternal IgG response, it did depend on the existence of a transient spike in IgG post-vaccination which is not inherent to IgG responses to all antigens. Therefore, these insights may not necessarily be extrapolated to other prenatal immunization programs. However, it is worth noting that our modeling framework is modular, allowing similar optimization routines to be carried out for other prenatal vaccines so long as there is sufficient data to inform simulations of the maternal antigen-specific IgG response kinetics.

From a precision medicine standpoint, the model-driven finding that IgG competes for placental transfer contingent upon subclass abundance, FcR expression, and Fc-FcR affinity provides a basis for improving vaccine formulations and dosing strategies to optimize vaccine-induced IgG transfer efficiency. The mathematical model devised in the current study provides a flexible platform to test these novel vaccination strategies given a set of user-defined inputs. Recent studies suggest that modulating vaccine parameters such as the nature of the target antigen, conjugate, number of doses, route of entry, and choice of adjuvant can fine-tune the pool of maternal vaccine-induced antibodies and the anti-infection capabilities of the IgG transferred to the newborn. For example, protein antigens tend to induce an IgG1 or IgG3 response, while polysaccharide antigens tend to induce an IgG2 response [62]; more recent studies suggest that the N-glycan profile of vaccine-induced IgG can be modulated by adjuvant, providing a means to fine-tune Fc receptor binding potential [51,52,63,64]. Using our model, we discovered the efficacy of maternal vaccines to transfer IgG to the fetus depended strongly on expression of FcRn in STBs and FcγRIIb in ECs. Our model predicted that readily tunable vaccine design choices such as timing of administration, dosage, and adjuvant are potential compensatory mechanisms to improve α-PT IgG transfer in patients with a deficiency in either FcRn in STBs or FcγRIIb in ECs. In order to make such recommendations, further research is needed to identify readily ascertainable biomarkers of placental FcR expression and/or transcytosis efficiency. As a more immediately actionable insight, we further uncovered a sensitivity of optimal immunization timing to volume parameters including maternal, fetal, and placental stromal volumes, which can be estimated by ultrasound imaging or alternative methods. Moreover, our model is able to quantitatively assess the effect of a perturbed maternal pool of antibodies due to various immunodeficiency disorders, autoimmune disorders, and other disorders that modify the maternal pool of antibodies and vaccine response [65]. As such, we envision this model as a tool to devise precision immunization approaches that account for maternal vaccination history, baseline pool of antibodies,

autoimmune disorders or other immunodeficiencies, and risk for preterm delivery or other pregnancy complications to determine the optimal immunization approach per patient.

While this model-driven investigation generated many intriguing insights, there are a few limitations to consider. First, the current model relies on the assumption that all IgG is mono-valent with a uniform glycan profile. This is an oversimplification as IgG glycosylation under-goes a dynamic shift during pregnancy [66], which can affect Fc receptor-mediated transfer [12,53]. Furthermore, in addition to monomeric IgG, IgG can potentially be transferred as antigen-antibody complexes, which is currently understudied and is not incorporated in our current model [16,34]. Second, there is a stark lack of data surrounding human placental development and dynamic fetal IgG levels throughout gestation; many model parameters were unavailable in the literature, necessitating *in silico* parameter optimization. Consequently, we assumed several model components to be constant, including maternal, fetal, and placental volumes and maternal IgG levels. In actuality, these features undergo rapid remodeling and growth across gestation and potentially fine-tune antibody transfer dynamics. Future iterations of the model may benefit from more physiologically accurate dynamics of the developing maternal-fetal interface. To facilitate such a model, further investigations detailing the dynamics of placental growth and FcR expression at the protein level are warranted.

Despite its limitations, a key strength of our computational approach is the ability to simulate placental transport dynamics which are prohibitively challenging to study *in vivo*. Using tightly integrated modeling and experimental methods, we probed placental transport across scales in time and space, enabling a more holistic view of the system compared to previous studies in the field. Traditionally, placental IgG transport efficiency is benchmarked by a simple ratio of maternal to umbilical cord IgG titer at birth, in part due to its ease of measurement and interpretability. Not only does this metric mask the nuanced impact of the maternal pool of antibodies, but also it insufficiently describes the highly dynamic and cumulative process of antibody transfer and is thus not capable of describing: (1) important concepts such as placental FcR limitation and antibody subclass competition, (2) the interplay between the maternal pool of antibodies and the vaccine-induced maternal antibodies, and (3) the role of complex placental functions that rapidly evolve throughout gestation, among other reasons. Our mathematical model challenges this metric as a true measure of "efficiency". With the advent and accessibility of state-of-the-art high-throughput measurements, it is increasingly feasible to probe placental function and define one or more quantitative measures of transfer efficiency that are conditioned on the maternal pool of antibodies and are linked with distinct model parameters that account for placental size, perfusion, and Fc receptor expression.

This study represents a major thrust forward in the field of maternal-neonatal immunology and paves the way towards next generation vaccination programs to fine-tune neonatal immunity against a broad range of pathogens. As current vaccination regimens are continually revised and several novel vaccines are currently in clinical trials, now is an opportune time to increase the prevalence of computational systems biology approaches in maternal-fetal medicine to improve outcomes for vulnerable newborns.

## Methods

### Mechanistic model of placental IgG transport

An ordinary differential equation (ODE) model was formulated to simulate mass transport of IgG1, IgG2, IgG3, and IgG4 across the maternal-fetal interface (S1 Appendix). A schematic of the model is given in Fig 1A. Fundamental model assumptions are as follows: (i) all processes obey the law of mass action; (ii) IgG binds FcRn within STB endosomes at an acidified pH and FcγRIIb at the EC surface [7,60,67,68]; (iii) binding with either receptor initiates transcytosis;

(iv) the effects of FcRn-mediated IgG recycling to the apical STB surface are negligible; (v) unbound IgG in STB and EC endosomes undergoes lysosomal degradation; (vi) all blood volumes, IgG subclass concentrations in maternal blood, and IgG-FcR affinities remain constant throughout gestation; (vii) rate parameters ($k_{up}$, $k_{trans}$, and $k_{deg}$) are equal in STB and EC layers and across IgG subclasses; (viii) stromal cells (e.g., Hofbauer cells, fibroblasts) do not interact with IgG or affect its transfer; (ix) the effect of Fab-antigen interactions is neglected such that all IgG is monovalent; (x) FcR expression increases parabolically across gestation informed by FcRn expression trends in rat placenta [31] (see S1 Appendix); (xi) STBs exclusively express FcRn and ECs express both FcRn and FcγRIIb; (xii) the effects of Fc or Fab glycosylation on FcR binding are neglected. All model development and corresponding *in silico* experiments were performed in MATLAB R2022a (Mathworks).

## Parameter determination

In total, 17 parameters were derived from literature and 11 were optimized. $K_D$ parameters were ascertained from Bruhns *et al*, where the authors determined the equilibrium association constants ($K_A$) experimentally by surface plasmon resonance (SPR) and we determined $K_D$ by the formula $K_D = 1/K_A$ [8,37]. We used these $K_D$ values to determine parameters $k_{on}$ and $k_{off}$ for each Fc receptor and IgG subclass pair. The value of $k_{off}$ for IgG1-FcRn was determined experimentally by Suzuki *et al* using SPR [38]. We assumed $k_{off}$ to be constant for all subclasses based on evidence from other studies comparing multiple subclasses [69–72]. We derived the value of $k_{on}$ for each subclass-FcR interaction from the formula $k_{on} = k_{off} / K_D$. The rate of IgG decay in the fetus ($\delta_{AB}$) was derived from the half-life of IgG1 measured in human infants by the formula $\delta_{AB} = \ln(2)/t_{1/2}$ [39]. Remaining parameter ranges were determined by calibrating predictions of IgG subclass levels in the fetus to data from Malek *et al*, where the authors quantified IgG subclass concentrations in umbilical cord blood samples by enzyme linked immunosorbent assay (ELISA) [32]. Initial conditions (maternal IgG subclass concentrations) were directly ascertained from [32].

We performed parameter estimation for 11 parameters using CaliPro, a flexible optimization software implemented in MATLAB and described by Joslyn *et al* [33]. A key strength of CaliPro is its ability to home in on a biologically relevant range of parameters that recapitulate the system dynamics rather than minimizing a simple cost function. Briefly, we implemented CaliPro using Latin Hypercube Sampling (LHS) from a physiologically plausible range of parameter values as determined by benchmarking studies found in the literature. We defined the pass criteria as model simulations within the upper standard deviation of the data set at the first time point, and are within 1.5 times the standard deviation at the final time point. Our termination criteria was set to 95% of model runs meeting the pass criteria.

## Global sensitivity analysis

Global sensitivity analysis was performed using LHS to sample the multi-dimensional parameter space followed by an orthogonalized partial least squares regression (OPLSR) framework, as described in [41,42]. OPLSR is a multivariate regression technique that predicts a single outcome (Y) from multiple predictors (X) and is especially suited to model systems with multicollinearity among predictors [73]. Briefly, numerous random perturbations to all *parameters* of the model were made simultaneously and resulted in individual model runs per random parameter set. We then leveraged OPLSR to build a model for an output of interest (Y) from the matrix of randomly perturbed parameter sets (X). Specifically, to produce the matrix of predictors, 1000 perturbed parameter sets were generated by LHS from a uniform distribution spanning the maximum likelihood range for each parameter previously obtained from

CaliPro. The number of parameter sets tested was fixed to the lowest sample size needed to achieve stable sensitivity analysis results. X is therefore a 1000 x 11 matrix (perturbations x parameters) and Y is a 1000 x 1 vector (perturbations x outcome). First, an OPLSR model was constructed using the matrix of 1000 permuted parameter sets to predict the total concentration of IgG in the fetus at 40 weeks gestation. A second OPLSR model was constructed in the same manner to predict entropy among IgG subclasses in the fetus at 40 weeks gestation, defined as:

$$P_{IgGi} = \frac{IgG_i}{\sum_{i=1}^{4} IgG_i}$$

$$E = -\sum_{i=1}^{4} P_{IgGi} \log(P_{IgGi})$$

Where E is entropy and $P_{IgGi}$ is the proportion of each subclass in the fetal blood. Model orthogonalization was performed such that the direction of maximum variance was aligned with latent variable 1 (LV1) [74]. A third OPLSR model was constructed to determine the sensitivity of the optimal $t_{vax}$ to each parameter.

OPLSR model significance was determined empirically by comparing its predictive power ($Q^2$) against randomly permuted models in a 5-fold cross validation scheme, where $Q^2$ is defined as:

$$Q^2 = 1 - \frac{\sum_i (y_i - f'(x_i))^2}{\sum_i (y_i - \bar{y})^2}$$

Where $x_i$ are the corresponding randomly perturbed parameter vectors in matrix X, $y_i$ is each individual outcome in vector Y, and $\bar{y}$ is the mean of all observations in Y.

## HUVEC transcytosis assay

**Antibody preparation.** Purified human polyclonal IgG1 and IgG4 (Abcam, AB90283, AB183266) were dialyzed in 1x PBS using Slide-A-Lyzer mini dialysis units with a 10K molecular weight cutoff (Invitrogen, 88404) to remove sodium azide. Dialysis units were incubated at 4°C for 2 hours, then dialysis buffer was changed and incubated again at 4°C for 2 hours. Protein was removed from the dialysis unit using a micropipette and quantified using a Nano-Drop Microvolume Spectrophotometer (ThermoFisher).

**Cell culture.** 6.5 mm diameter Transwell inserts (Corning, CLS3422) coated in 1.4% gelatin were seeded with primary human umbilical vein endothelial cells (HUVEC) (Promocell C-12200, kindly provided by Dan Gioeli at the University of Virginia) at a density of 1,500 cells/mm$^2$ (50,000 cells/insert). Cells were cultured in complete endothelial cell growth media (Promocell, C-22110) and incubated at 37°C and 5% CO$_2$. After 24 hours, the cells were washed in sterile PBS and the media was changed.

**Transcytosis assay.** After 48 hours, the cells were washed in sterile PBS and media containing purified human IgG1 and IgG4 was added to the apical chamber. Cells were incubated at 37°C and 5% CO$_2$ for 120 minutes. IgG1 and IgG4 in the basolateral media were quantified using human ELISA IgG1 and IgG4 quantification kits (Invitrogen, EHIGG1 and BMS2095) according to the manufacturer's protocol. All conditions were run in triplicate and ELISA measurements were run in duplicate. A 5-parameter logistic standard curve was generated for each assay. Absorbance was read at 450 nm with a correction of 590 nm using an Optima microplate reader within 1 hour of completing the assay. Statistical significance was

determined with a one-tailed two-sample t-test comparing each sample containing IgG1 and IgG4 to the sample with IgG4 but without IgG1.

To confirm monolayer formation, membrane permeability was quantified for each assay by addition of FITC-dextran (4000 MW, Sigma Aldrich 46944) (0.5 mg/ml) to the apical chamber of each insert. FITC-dextran added to inserts without gelatin coating or HUVEC served as a negative control. The basolateral media was sampled after 120 minutes, and fluorescence was read using the Optima microplate reader. Monolayer permeability (%) was defined as:

$$Permeability = \frac{[basolat.FITC - dextran]_{HUVEC}}{[basolat.FITC - dextran]_{blank}} \times 100$$

Samples with FITC-dextran permeability greater than 3 standard deviations from the mean were excluded based on faulty monolayer formation (S5 Fig).

**Mechanistic model of HUVEC transcytosis *in vitro*.** To simulate the dynamics of IgG1 and IgG4 receptor-mediated transcytosis in HUVEC, a kinetic-dynamic model of FcRn-IgG binding and transcytosis in endothelial cells was formulated:

$$\frac{dIgG1}{dt} = -k_{on,1}[IgG1][FcRn] + k_{off,1}[C_1] \tag{1}$$

$$\frac{dIgG4}{dt} = -k_{on,4}[IgG4][FcRn] + k_{off,4}[C_4] \tag{2}$$

$$\frac{dFcRn}{dt} = -[FcRn]\left(k_{on,1}[IgG1] + k_{on,4}[IgG4]\right) + \left(k_{off,1} + k_{trans}^{HUVEC}\right)[C_1] + \left(k_{off,4} + k_{trans}^{HUVEC}\right)[C_4] \tag{3}$$

$$\frac{dC_1}{dt} = k_{on,1}[IgG1][FcRn] - \left(k_{off,1} + k_{trans}^{HUVEC}\right)[C_1] \tag{4}$$

$$\frac{dC_4}{dt} = k_{on,4}[IgG4][FcRn] - \left(k_{off,4} + k_{trans}^{HUVEC}\right)[C_4] \tag{5}$$

$$\frac{dIgG1_T}{dt} = k_{trans}^{HUVEC}[C_1] \tag{6}$$

$$\frac{dIgG4_T}{dt} = k_{trans}^{HUVEC}[C_4] \tag{7}$$

Where [IgG1] and [IgG4] are apical concentrations of IgG1 and IgG4, [FcRn] is the concentration of FcRn expression in HUVEC, [$C_1$] and [$C_4$] are the concentration of bound IgG-FcRn complexes, $k_{trans}^{HUVEC}$ is the rate of transcytosis, $k_{on}$ is the forward rate of reaction, and $k_{off}$ is the reverse rate of reaction (Table 2). The model parameters were optimized by fitting to dynamic data of IgG4 transcytosis in HUVECs collected at 15, 30, 60, and 120 minutes ([IgG4$_{apical}$] = 0.5 and 0.125 mg/ml) (S5 Fig). To simplify this model and reveal the effect of subclass-subclass competition in Fc receptor-mediated transcytosis, we derived the closed form of equation (5). We assumed that IgG1 and IgG4 are present in excess of FcRn and the system is at quasi-steady state such that

$$\frac{dC_1}{dt} = \frac{dC_4}{dt} = 0 \tag{8}$$

Where C is the concentration of IgG-FcRn complexes. Under these assumptions, Eqs (4) and

(5) can be rearranged as

$$\frac{k_{on,1}[IgG1][FcRn]}{k_{on,1}[IgG1][FcRn](k_{off,1} + k_{trans}^{HUVEC})} = [C_1] \qquad (9)$$

$$\frac{k_{on,4}[IgG4][FcRn]}{k_{on,4}[IgG4][FcRn](k_{off,4} + k_{trans}^{HUVEC})} = [C_4] \qquad (10)$$

The assumption is then made that the total concentration of receptor does not change such that

$$[FcRn] = [FcRn_0] - [C_1] - [C_4] \qquad (11)$$

Substituting (11) into (10):

$$\frac{k_{on,4}[IgG4]([FcRn_0] - [C_1] - [C_4])}{k_{on,4}[IgG4][FcRn](k_{off,4} + k_{trans}^{HUVEC})} = [C_4] \qquad (12)$$

And (9) into (12):

$$\frac{k_{on,4}[IgG4]([FcRn_0] - \frac{k_{on,1}[IgG1][FcRn]}{k_{on,1}[IgG1][FcRn](k_{off,1} + k_{trans}^{HUVEC})} - [C_4])}{k_{on,4}[IgG4][FcRn](k_{off,4} + k_{trans}^{HUVEC})} = [C_4] \qquad (13)$$

Solving (13) for [C_4] yields the closed-form solution for IgG4-FcRn complex formation [75]:

$$C_4 = \frac{FcRn_0[IgG4]}{K_{D,4}(1 + \frac{[IgG4]}{K_{D,4}} + \frac{[IgG1]}{K_{D,1}})} \qquad (14)$$

Which quantifies the effect of competition between two IgG subclasses undergoing transcytosis in the presence of one Fc receptor.

## Pertussis (Tdap) immunization simulations

To simulate maternal Tdap immunization, a model of plasma cell activation and IgG secretion in response to antigen stimulus was adapted from a previous study by White *et al* in the context of malaria [45]. The model consisted of 8 parameters, of which 4 were fitted parameters determined by White *et al*, 1 was determined experimentally by Dari *et al*, and 3 were optimized in this study [47]. Parameters were tuned to fit previously published dynamic data of the maternal IgG response to Tdap booster vaccination [46] (Table 3). We assumed several parameters which depend on the host environment—namely, systemic cytokine concentrations and availability of FcRn for IgG recycling and half-life extension—do not depend on antigen-specificity and thus are conserved between malaria and pertussis contexts [76,77]. Such parameters were preserved from White *et al* and include the proportion of long-lived antibody secreting cells ($L_{ASC}$) to short-lived antibody secreting cells ($S_{ASC}$), the lifespan of $L_{ASC}$ and $S_{ASC}$, and the antibody decay rate [45]. On the other hand, parameters which depend on the plasma cell niche and context of the specific antigenic challenge (e.g., rates of plasma cell proliferation, antibody secretion, and the antigen decay rate) were optimized to fit titers of anti-pertussis toxin IgG ($\alpha$-PT IgG) in women of childbearing age following a booster Tdap vaccination [46] (Table 3). Other important model assumptions include: (i) IgG has a half-life of 31 days in the fetus; (ii) the dynamics of B cell activation—including antigen presentation by follicular dendritic cells and cytokine secretion by T cells—and the contribution of memory B cells are neglected; (iii) mothers have been previously exposed to pertussis toxin (i.e., this is a

booster vaccine); (iv) all mothers have the same dynamic response to immunization; (v) immunization resulted in equal parts IgG1, IgG2, IgG3, and IgG4 with uniform glycosylation profile and immunoglobulin isotype switching was neglected; (vi) $L_{ASC}$s and $S_{ASC}$s secrete IgG with identical subclass and glycosylation profile; (vii) the peak antibody titer is between 14 and 28 days post-immunization; (viii) the IgG response to pertussis is the same in pregnant and non-pregnant individuals [78]. Maternal immunization was simulated by stimulating the maternal compartment with antigen at a time $t_{vax}$ (given in weeks gestational age) to induce an antibody response, which directly fed into the maternal compartment of the placental IgG transport model. To determine the optimal $t_{vax}$ corresponding to maximum pertussis toxin-specific IgG transfer to the fetus, iterative simulations were performed over a range of $t_{vax}$ spanning 10–38 weeks gestational age. Sensitivity of the optimal $t_{vax}$ was determined by parameter perturbation and subsequent OPLSR model construction, as described above.

## Supporting information

**S1 Fig. Fc receptors are incorporated as dynamic offline variables.** $FcRn_{STB}$, $FcRn_{EC}$, and $Fc\gamma RIIb_{EC}$ were modeled as second order polynomials based on trends observed in rat placenta in a previous study by Wang *et al* [31]. The fit for $FcRn_{STB}$ is shown here as an example. The trend of $FcRn_{STB}$ protein expression levels in rat placenta was scaled to the length of human gestation. The concentration of $FcRn_{STB}$ at full term was an optimized parameter. Additional constraints to achieve similar convex dynamics as observed by Wang *et al.* [31] are shown in blue. See Equation 13 (S1 Appendix) for more details.
(PDF)

**S2 Fig. Mechanistic model of placental transfer recapitulates bulk IgG transfer.** Bulk IgG in umbilical cord blood measured using cordocentesis in a cohort spanning gestational ages is shown as the cohort mean and standard deviation in grey circles. Simulated total fetal IgG and IgG subclasses across gestation are overlaid as solid curve. The curves for IgG3 and IgG4 are overlapping.
(PDF)

**S3 Fig. Preferential expression of *FCGR2B* over *FCGRT* by placental endothelial cells increases across gestation.** (A-C) The scatter plots show the expression of *FCGR2B* and *FCGRT* in placental ECs in scRNA-seq data sets from three gestational time points: (A) first trimester, (B) early third trimester, and (C) late third trimester. Each dot corresponds to a single cell. Line y = x is indicated on each panel. In panel (A), a swarmchart of FcRn expression is included as an inlet to demonstrate the abundance of cells more clearly at the origin. (D) The trend of FcRn and FcγRIIb expression is shown across gestation as the percentage of ECs expressing FcγRIIb (circles) and FcRn (triangles) at each time point.
(PDF)

**S4 Fig. IgG subclasses compete for FcγRIIb-mediated transcytosis in the placental mechanistic model.** (A) The bar plot represents the F:M ratio at parturition for a term pregnancy from independent simulations in the compartmental placental model with each subclass isolated (solid fill) or mixed with other IgG subclasses ("Competition", patterned fill). (B-C) The F:M ratio at parturition for a term pregnancy is shown for each subclass in isolation (B) or mixed with other subclasses (C) as in (A) over a range of FcγRIIb expression. (D) The difference (delta) in F:M ratio at parturition for a term pregnancy between isolation and competition conditions is shown over a range of FcγRIIb expression. In (B-D), the optimized range of FcγRIIb expression is indicated by the shaded region.
(PDF)

**S5 Fig. Transwell model parameters were fit to longitudinal *in vitro* data. (A)** Longitudinal dynamic data from IgG transcytosis measured *in vitro* in a HUVEC model are shown as the mean and standard deviation of 3 replicates. The overlaid curves show corresponding model simulations described in Methods. **(B)** The FITC-dextran permeability measurements demonstrates the criteria for rejecting a data point on the basis of faulty monolayer formation.
(PDF)

**S6 Fig. The optimal immunization window during pregnancy depends on the existence of a transient spike in antigen-specific IgG.** The existence of a transient spike in α-PT IgG is driven by parameter ρ, the proportion of short-lived antibody secreting cells. Three cases are shown within the range of optimized values of ρ reported in White *et al.*[45] When ρ = 68% (top), the antibody response is driven by a larger proportion of long-lived antibody secreting cells resulting in a persistently high level of anti-PT IgG. Consequently, optimal immunization times were earlier in gestation for the majority of gestational age groups. When ρ = 82% (middle), the antibody response peaks and gradually declines and optimal immunization times were between 10–19 weeks gestation. When ρ = 96% (bottom), the spike in maternal IgG levels is highly transient and optimal immunization time scales with gestational age.
(PDF)

**S7 Fig. The magnitude of the maternal α-PT IgG response does not affect the predicted optimal immunization window.** Three models of the maternal α-PT IgG response to Tdap immunization were fit to the median (squares), upper (circles), and lower (triangles) 95% confidence intervals reported by van der Lee *et al.* [47] The magnitude of the maternal α-PT IgG spike affected the level of α-PT IgG present in the fetus at the time of delivery but did not impact timing of the optimal immunization window.
(PDF)

**S1 Appendix. Placental IgG transfer model equations.**
(PDF)

**S2 Appendix. Tdap immunization model equations.**
(PDF)

**S1 Data. IgG transcytosis in endothelial cells (*in vitro* data).**
(XLSX)

## Acknowledgments

We thank the members of the Dolatshahi lab, Drs. Donald J. Dudley and Loren D. Erickson for helpful discussion and feedback. We would also like to thank Dr. Daniel Gioeli and Devin Roller for providing cell lines and support with Transwell assays. We would like to acknowledge Drs. Yuk Ming Dennis Lo, Peiyong Jiang, and Xi Hu from the Chinese University of Hong Kong for sharing their published single cell transcriptomic data of placental tissue. We acknowledge Research Computing at The University of Virginia for providing computational resources and technical support that have contributed to the results reported within this publication. URL: https://rc.virginia.edu. The content is solely the responsibility of the authors and does not necessarily represent the official views of the funding agencies and the National Institutes of Health.

## Author Contributions

**Conceptualization:** Remziye E. Wessel, Sepideh Dolatshahi.

**Data curation:** Remziye E. Wessel.

**Formal analysis:** Remziye E. Wessel, Sepideh Dolatshahi.

**Funding acquisition:** Sepideh Dolatshahi.

**Investigation:** Sepideh Dolatshahi.

**Resources:** Sepideh Dolatshahi.

**Software:** Remziye E. Wessel.

**Supervision:** Sepideh Dolatshahi.

**Validation:** Remziye E. Wessel, Sepideh Dolatshahi.

**Visualization:** Remziye E. Wessel, Sepideh Dolatshahi.

**Writing – original draft:** Remziye E. Wessel, Sepideh Dolatshahi.

**Writing – review & editing:** Remziye E. Wessel, Sepideh Dolatshahi.

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
