## [Decision Letter · Decision Letter 0]

1 Jul 2023

Dear Dr. Dolatshahi,

Thank you very much for submitting your manuscript "Quantitative mechanistic model reveals key determinants of placental IgG transfer and informs prenatal immunization strategies" for consideration at PLOS Computational Biology.

As with all papers reviewed by the journal, your manuscript was reviewed by members of the editorial board and by several independent reviewers. In light of the reviews (below this email), we would like to invite the resubmission of a significantly-revised version that takes into account the reviewers' comments.

We cannot make any decision about publication until we have seen the revised manuscript and your response to the reviewers' comments. Your revised manuscript is also likely to be sent to reviewers for further evaluation.

Sincerely,

Rustom Antia

Academic Editor

PLOS Computational Biology

Kiran Patil

Section Editor

PLOS Computational Biology

Reviewer's Responses to Questions

**Comments to the Authors:**

Reviewer #1: Erdogan and Dolatshahi explored how maternal antibodies are transferred to the fetus in utero using a mechanistic model. Specifically, they constructed a model for FcRn- and FcgRIIb-mediated IgG transport on STBs and ECs and analyzed the sensitivity of each parameter. They also explored the role of FcgRIIb in IgG transfer across fetal ECs as well as evidence of subclass competition. Finally, they applied the model to identify techniques for optimizing vaccination scheduling. Studying placental antibody transfer is highly relevant to maternal health, and this study has the potential to impact vaccine design and timing. However, I have several concerns, about assumptions from the model and questions regarding what kind of actionable insights can be derived from it, that would need to be addressed before publication.

Major issues:

The authors need to clearly define which observations are insights derived from the present study versus prior literature. For example, the propensity for transfer of each subclass is derived from previous studies, and so steering vaccine subclass production, as discussed at length in the discussion, does not seem like an implication of this work. The dynamical nature of the model does seem like a unique contribution that the authors could emphasize more.

The practical implications of this model should be clarified. This work pointed out the factors that are positively or negatively associated with earlier or later optimal vaccination time, but many of them are not actionable. E.g., one can’t change their endosomal volume, FcR expression in specific cells, etc. Similarly, this work suggests that the model can be used for personalized medicine. What measurements would be made to calibrate the model for specific individuals?

What is the rationale for only including FcgRIIb on ECs in the model? The authors have pointed out in Fig. 3 that the ECs express a mixture of FcRn and FcgRIIb, and the possible involvement of other FcgR (such as RIIIa). It seems strange that the FcRn/FcgRIIb model was only proposed in Fig. 3 but not the original model. Given these other possibilities are not ruled out as inconsistent with the data, it is not possible to determine whether the resulting model is the only one consistent with the evidence.

Observing a distinct optimum in vaccination timing seems very dependent on vaccination leading to a highly transient spike in antibodies which is, in turn, dependent on the proportion of short-lived ASCs. 96% was used in this study, but the cited study observes 68%–95%. Given the importance of this parameter, it should be discussed.

The authors treat FcgRIIb and FcRn identically aside from their differing affinities at neutral pH. However, FcRn is highly pH-dependent, leading to its unique role in endosomal transport. Is there evidence to justify handling the endosomal trafficking for each of these receptors in the same way?

Minor issues:

In Fig. 1A, is k_trans the same as k_t in the table? Keep the notation consistent.

Whether the model used in Fig. 4 for HUVEC is the same or different from the original one should be clearly stated.

In Fig. 5/Table 3, are decay rates d or δ? Keep the notation consistent.

In Table 1, K_D were listed as parameters, while the equations used k_on and k_off. How were the latter derived?

Fig. 3 caption, line 211, did you mean “Statistical significance in (A-B)”?

Fig. S3 is an interesting result. Consider including it in Fig. 3.

Fig. 4A, I wouldn’t say the effect is “proportional” in line 221. It is hard to say there is a specific relation like this with just two examples.

Is Fig. 4A the same as Fig. S4A except only showing IgG1 and IgG4? What is the composition of subclasses in “Competition”? Is the competition among two subclasses or four?

In Fig. 4D, it’d be helpful to mark where [IgG4_apical] > 0.33 is.

Fig. 6C: would it make more sense to draw the thicker line (cyan or blue) on the x-axes?

In Fig. 1B, data were overlaid on the simulated levels. Don’t see a description or a citation on the details of the data used.

Did Fig. 3a and 3b use the same data? If so, why was “< 16 weeks” not included in Fig. 3a?

In Fig. 3b, are “< 12 weeks” samples in line 174 the same as “< 16 weeks” in the figure? Are the “38 weeks” samples in line 176 the same as “> 37 weeks”?

In lines 224-226, the rationale for choosing IgG1 and IgG4 should have citations.

For HUVEC in the Transwell assays, are the “three independent replicates” biological or technical?

Reviewer #2: In this manuscript, the authors have modelled transplacental antibody transfer to optimize immunization of newborns via vaccination of mothers. The authors have identified expression of FcγRIIb on the endothelial cells to be a key limiting factor shaping the observed selectivity of different IgG subclasses. Combining this model with a vaccination model, the authors have identified an optimal gestational age range for vaccination of mothers to maximize neonatal immunity. Overall, I found this manuscript very well-written. Below are some minor comments, which should be addressed before publication.

Minor comments:

How can the values of FcRn and FcγRIIb expression be optimized as reported in Table 1, I thought they are variables in the model? According to the complete model equations in the SI, parameters a,b, and c needs to be estimated for the expression of FcRn and FcγRIIb.

In table 1, for the parameters taken from published research, mainly from references 29 and 31, it will be good to know how these parameter values were calculated. If the values were directly taken from these references, please describe how they calculated the values. I suggest describing this in the Methods section.

Please mention how many parameters were estimated and how many parameters were taken from literature in total.

In Fig 2, it is not clear how to interpret different points in fig A and C. Please explain briefly how OPLSR works.

**Have the authors made all data and (if applicable) computational code underlying the findings in their manuscript fully available?**

Reviewer #1: Yes

Reviewer #2: Yes

PLOS authors have the option to publish the peer review history of their article (what does this mean?). If published, this will include your full peer review and any attached files.

Reviewer #1: No

Reviewer #2: No
---

## [Decision Letter · Decision Letter 1]

12 Oct 2023

Dear Dr. Dolatshahi,

We are pleased to inform you that your manuscript 'Quantitative mechanistic model reveals key determinants of placental IgG transfer and informs prenatal immunization strategies' has been provisionally accepted for publication in PLOS Computational Biology.

Best regards,

Rustom Antia

Academic Editor

PLOS Computational Biology

Kiran Patil

Section Editor

PLOS Computational Biology

Reviewer's Responses to Questions

**Comments to the Authors:**

Reviewer #2: All the comments have been addressed properly.

**Have the authors made all data and (if applicable) computational code underlying the findings in their manuscript fully available?**

Reviewer #2: Yes

PLOS authors have the option to publish the peer review history of their article (what does this mean?). If published, this will include your full peer review and any attached files.

Reviewer #2: No

---

## [Editor Report · Acceptance letter]

26 Oct 2023

PCOMPBIOL-D-23-00617R1 

Quantitative mechanistic model reveals key determinants of placental IgG transfer and informs prenatal immunization strategies

Dear Dr Dolatshahi,

I am pleased to inform you that your manuscript has been formally accepted for publication in PLOS Computational Biology. Your manuscript is now with our production department and you will be notified of the publication date in due course.

With kind regards,

Anita Estes
